# The Influence of Antibiotic Resistance on Innate Immune Responses to *Staphylococcus aureus* Infection

**DOI:** 10.3390/antibiotics11050542

**Published:** 2022-04-19

**Authors:** Nazneen Jahan, Timothy Patton, Meredith O’Keeffe

**Affiliations:** 1Department of Biochemistry and Molecular Biology, Biomedicine Discovery Institute, Monash University, Clayton, VIC 3800, Australia; nazneen.jahan@monash.edu; 2Department of Microbiology and Immunology, Peter Doherty Institute for Infection and Immunity, The University of Melbourne, Melbourne, VIC 3000, Australia; timothy.patton@unimelb.edu.au; 3Centre for Innate Immunity and Infectious Disease, Hudson Institute of Medical Research, Clayton, VIC 3168, Australia

**Keywords:** *Staphylococcus aureus*, antibiotic resistance, innate immunity

## Abstract

*Staphylococcus aureus* (*S. aureus*) causes a broad range of infections and is associated with significant morbidity and mortality. *S. aureus* produces a diverse range of cellular and extracellular factors responsible for its invasiveness and ability to resist immune attack. In recent years, increasing resistance to last-line anti-staphylococcal antibiotics daptomycin and vancomycin has been observed. Resistant strains of *S. aureus* are highly efficient in invading a variety of professional and nonprofessional phagocytes and are able to survive inside host cells. Eliciting immune protection against antibiotic-resistant *S. aureus* infection is a global challenge, requiring both innate and adaptive immune effector mechanisms. Dendritic cells (DC), which sit at the interface between innate and adaptive immune responses, are central to the induction of immune protection against *S. aureus*. However, it has been observed that *S. aureus* has the capacity to develop further antibiotic resistance and acquire increased resistance to immunological recognition by the innate immune system. In this article, we review the strategies utilised by *S. aureus* to circumvent antibiotic and innate immune responses, especially the interaction between *S. aureus* and DC, focusing on how this relationship is perturbed with the development of antibiotic resistance.

## 1. Introduction

*S. aureus* is mainly found living on the skin and mucus membranes of the anterior nares of most healthy human populations (20–80%) [1,2]. Under normal circumstances, *S. aureus* maintains a symbiotic relationship with the host but becomes virulent in deeper tissue sites accessed through trauma or parenteral route of infection [3]. Staphylococcal bacteraemia is the leading cause of human morbidity and mortality in hospital settings [4]. It is responsible for various diseases ranging from mild skin infections (impetigo, folliculitis) to more invasive infections (wound infections, endocarditis, osteomyelitis) and toxin-mediated diseases (toxic shock syndrome, scaled skin syndrome) [5]. Pathogenic strains of *S. aureus* are also associated with respiratory tract infections that range from asymptomatic colonisation to necrotizing pneumonia [6]. This necrotizing pneumonia is a progressive form of pneumonia causing respiratory distress, pleural effusion, haemoptysis, and leukopenia [7]. Disease progression in staphylococcal pneumonia is mainly mediated by the excessive and dysregulated host inflammatory response to infection causing lung injury [8,9].

*S. aureus* infection is of high socioeconomic burden in both developed and developing nations [10]. The emergence of ‘super bugs’ that have shown resistance to multiple antibiotics, leading to increased reliance on last-line antibiotics such as vancomycin and daptomycin, and that have shown an ability to simultaneously avoid host innate immunity [4,11] further complicates treatment strategies of *S. aureus* infections. Infections caused by daptomycin-resistant *S. aureus* are particularly more prolonged and difficult to treat [4,11,12]. Analyses of antibiotic-resistant *S. aureus*-host innate immune interactions may help to discover the fine details of pathogenesis of infections involving these bacterial strains and to develop effective therapeutic options.

## 2. Innate Sensing of *S. aureus*

The innate immune response is the first line of defence against invading pathogens. The innate immune system is comprised of diverse cells, proteins, and receptor systems that recognise pathogens and initiate a sequence of events that result in the production and secretion of inflammatory mediators [12]. The subsequent recruitment and activation of phagocytic cells to the infection site leads to a reduction in pathogen load and the downstream initiation of adaptive immunity [13,14]. Phagocytic cells of the innate immune system include neutrophils, macrophages, monocytes, and dendritic cells (DC). These cells are equipped to recognise evolutionarily conserved pathogen-associated molecular patterns (PAMPs) or damage-associated molecular patterns (DAMPs) through overlapping and cell-type-specific expression of pattern recognition receptors (PRRs) [15]. Major subtypes of PRR that have direct effects on initiating innate immunity are present on the cell surface, intracellular compartments, and in the cytosol. These include the toll-like receptors (TLRs), the retinoic acid inducible gene I-like receptors (RLRs), nucleotide-binding oligomerisation domain (NOD)-like receptors (NLR), C-type lectin receptors (CLR), and DNA sensing molecules such as cyclic GMP-AMP (cGAMP) synthase (cGAS) and stimulator of interferon genes (STING) (Table 1) [16,17].

### 2.1. TLRs

TLRs are the best-characterised pattern recognition receptors [16]. They have greatly advanced our understanding of how the body senses pathogen invasion, triggers innate immune responses, and primes antigen-specific adaptive immunity [15,18]. All TLRs except TLR3 utilise the MyD88-dependent signalling pathway that leads to nuclear factor-κB (NF-κB) activation and inflammatory responses. MyD88-deficient macrophages release much less cytokine than wild-type macrophages after staphylococcal infection [19]. Moreover, people with MyD88 deficiency have a high incidence of recurrent invasive and noninvasive *S. aureus* infections [41], highlighting the importance of TLR signalling for immunity to this bacterial strain. The roles of different TLRs in sensing PAMPS of *S. aureus* are outlined in this review (Table 1). TLR4, which classically recognises endotoxin, is not included in Table 1 since it is likely not involved in sensing of gram-positive bacteria such as *S. aureus*. However, recently, it has been demonstrated that *S. aureus* virulence factor Phenol-soluble modulin (PSM) α1–α3 inhibited HMGB1-induced NF-κB activation by binding with TLR4 in HEK-Blue hTLR4 and THP-1 cells [42]. Whether this interaction is definitively endotoxin-free remains to be elucidated.

TLR2 can recognise a wide range of PAMPs derived from bacteria, fungi, parasites, and viruses [20] and generally forms heterodimers with TLR1 or TLR6 [18]. TLR2 is considered critical for both systemic and localised *S. aureus* infection as it is highly expressed on resident macrophages and recruited neutrophils and monocytes [21]. Studies on TLR2^−/−^ mice have shown that TLR2 is required for survival, abundant cytokine secretion, and effective clearance of *S. aureus* from the brain [22]. Even though TLR2 has not been reported to function as a phagocytic receptor, its absence is reported to cause a decrease in *S. aureus* phagocytosis [43,44,45]. Although others have reported that bacterial internalisation was not affected in the absence of the MyD88 adaptor in macrophages, suggesting that any role for TLR2 in facilitating bacterial uptake is independent of MyD88-dependent signalling [46].

TLR2/TLR6 heterodimers recognise diacylated lipopeptides and peptidoglycans from *S. aureus* [18]. Lipoteichoic acid (LTA) is a major outer cell wall component of *S. aureus*. LTA is thought to be recognised by TLR2 in cooperation with CD36 [47]. Indeed, loss of response to LTA via inborn defects in signalling downstream of TLR2 has been shown to lead to susceptibility to severe staphylococcal disease [48]. Of interest, antibodies to LTA can protect against severe staphylococcal disease in individuals with defects in the TLR2 signalling pathway [48].

Endosomal TLR7, 8 (human), 9, and 13 (murine only) sense the presence of single-stranded RNA (ssRNA), DNA, and ribosomal RNA (rRNA), respectively. Studies have shown that single deficiencies in TLR7, 9, or 13 have little or no effect in host defences in *S. aureus* infection models [23]. However, the combined deficiency of TLR7, 9, and 13 in macrophages resulted in the complete inhibition of interleukin (IL)-12 p70 and interferon (IFN)-β production, and increased susceptibility to both cutaneous and systemic *S. aureus* infection [23]. A strong TLR9-mediated Type I IFN response to DNA containing unmethylated CpG motifs of *S. aureus* was demonstrated in murine bone-marrow-derived DC [27]. In human monocytes, TLR7 and TLR8 can sense S. aureus tRNA and ssRNA degradation products, leading to IFN-β production [24,25,26]. Recently, it has been shown that TLRs 3, 7, and 9 all contribute to the sensing of *S. aureus*-released microvesicles by RAW 264.7 cells [24]. Overall, endosomal sensing of *S. aureus* nucleic acids does appear to play a role in innate sensing of *S. aureus* but the reliance on a single endosomal TLR is less clear, with likely differences between different cell types.

### 2.2. C-Type Lectin Receptor

C-type lectin receptors (CLRs) are a heterogenous group of transmembrane proteins, many of which are expressed on myeloid cells [49]. CLRs possess one or more carbohydrate recognition domains (CRDs) which mediate binding to its carbohydrate ligand. In addition, soluble/circulating CLRs exist that bind an array of carbohydrate patterns on pathogen surfaces [28].

The C-type lectin (CLEC) receptor langerin (CD207, also called CLEC4K) is expressed on Langerhans cells of the skin and subsets of conventional DC in mouse and human [50]. Human CD207 binds β-*N*-acetylglucosamine (GlcNAc) of teichoic acid of the *S. aureus* cell wall (WTA), inducing activation and innate cytokine responses by the DC subsets [51]. Certain strains of *S. aureus* express structurally different WTA, which is instead recognised by macrophage galactose-type lectin receptor (MGL; CD301), expressed by macrophages and DC, also inducing proinflammatory immune responses [52].

Mannose binding lectin (MBL) is a serum glycoprotein of humans and animals that is involved in opsonisation of complement on the surface of *S. aureus* through binding to carbohydrate linkages of WTA [53,54]. In humans, low levels of MBL and/or MBL2 polymorphism are responsible for increased susceptibility to *S. aureus* bacteraemia [55]. The MBL–WTA interaction is a particularly important innate defence mechanism against *S. aureus* infection in infants, which becomes redundant upon induction of antibodies against WTA [56]. Mice lacking MBL exhibit reduced proinflammatory cytokine production during *S. aureus* infection, indicating a role for MBL in regulating cytokine release [29]. MBL complexes with TLR2 within phagosomes and the cooperation amplifies the host response to *S. aureus* [30].

Many other CLRs have been implicated in uptake and recognition of bacteria [57]. It is likely that other CLRs interact with *S. aureus*, potentially dependent on host cell type and differences in lipid moieties present in different bacterial strains or cell wall changes induced upon acquisition of antibiotic resistance.

### 2.3. Cytosolic Sensors

#### 2.3.1. NOD-like Receptors (NLR)

NLRs are located in the cytosol and respond to a range of both PAMPs and DAMPs [31]. They are multidomain proteins composed of a variable N-terminal effector region and C-terminal leucine-rich repeats (LRRs) that sense PAMPs [32]. N-terminal effector regions consist of caspase recruitment domain (CARD), pyrin domain (PYD), acidic domain, and a centrally located NOD domain. Based on their N-terminal domain, NLRs are divided into four subfamilies: NLRA, NLRB, NLRC, and NLRP [31]. The activated NLRs show various functions that can be divided into four broad categories: inflammasome formation, signalling transduction, transcription activation, and autophagy [58].

Mice deficient in the NLRC family member NOD2 have increased susceptibility to *S. aureus* infection and a significantly higher bacterial tissue burden. Their reduced viability is due in part to loss of NOD2 leading to defective neutrophil phagocytosis and reduced proinflammatory cytokines IL-1β and IL-6 and/or antimicrobial peptides, e.g., β-defensin production, against *S. aureus* infection [31,33].

*S. aureus* α, β-, and γ- hemolysins; Panton–Valentine leucocidin (PVL) [59]; and leucocidin A/B [60] can activate the NLRP3 inflammasome in monocytes and macrophages resulting in maturation and secretion of IL-1β and IL-18 [61,62,63]. In human macrophages, the NLRP7 inflammasome is induced in response to microbial acylated lipopeptides and promotes ASC-dependent caspase-1 activation, IL-1β and IL-18 maturation, and restriction of intracellular bacterial replication [64].

#### 2.3.2. RIG-I-like Receptors Family

The retinoic acid-inducible gene I (RIG-I)-like receptors (RLRs) family consists of three DExD/H box RNA helicases located in the cytosol: retinoic acid-inducible gene (RIG-I), melanoma differentiation-associated gene 5 (MDA-5), and laboratory of genetics and physiology-2 (LGP-2) [65,66]. Both RIG-I and MDA-5 have tandem N-terminal caspase activation and recruitment domains (CARDs), a DExD/H box RNA helicase domain, and a C-terminal repressor domain (RD). Unlike RIG-I and MDA-5, LGP-2 contains only the RNA helicase domain and acts as a negative regulator of the other RLRs [67,68,69,70,71]. During infection, RIG-I dimerizes in an ATP-dependent manner [72] and the activated multimeric form of RIG-I or MDA5 then interacts with the downstream adaptor protein mitochondrial antiviral signalling protein (MAVS, also known as CARDIF and IPS-1) [36,68,73]. Ligation of RIG-I or MDA5 with MAVS activates the IKK-related kinase, TANK binding kinase 1 (TBK1), which further activates interferon regulatory factor (IRF)3/IRF7, resulting in the transcription of type I interferons [73,74]. MAVS also activates the NF-κB pathway through recruitment of TRADD, FADD, caspase-8, and caspase-10 [75,76].

RLRs are key sensors of virus infection that recognises intracellular RNA introduced to the cytosol [65,77]. Although RIG-I is known to recognise viral RNA, several studies have indicated the role of RIG-1 in the detection of bacterial RNA or DNA in a host cell type or pathogen specific manner [34,78,79,80,81,82]. Recently, it has been demonstrated that a human microglial cell line and primary astrocytes express RIG-I and stimulate IFN production via RIG-I-dependent signalling in the presence of *S. aureus* genomic DNA [34]. In another study, it was observed that the levels of MDA5 and RIG-1 mRNA were significantly increased in septicaemia patients infected with *S. aureus* when compared with healthy controls—whether there is a link to protective immunity during septicaemia is unclear [83].

#### 2.3.3. cGAS–STING Pathway

Cytosolic DNA sensing by the cyclic GMP–AMP synthase (cGAS)–stimulator of interferon genes (STING) pathway is implicated in host immune defence mechanisms. cGAS senses viral, bacterial, or host dsDNA aberrantly localised in the cytosol [84] and promotes cGAS oligomerisation and activation [85,86]. Activated cGAS catalyses the formation of cyclic dinucleotides (CDN) from ATP and GTP to the CDN 2′3′-cGAMP [87]. CDNs such as cyclic di-AMP or cyclic di-GMP can also be produced from bacteria and activate STING directly [36]. CDNs bind to the endoplasmic reticulum (ER)-localised STING, which promotes STING dimerization and translocation from the ER to golgi compartments [36,37] where STING recruits and activates TBK1 and nuclear translocation of the transcription factor IRF3) and to a lesser extent, NF-κB) [38], leading to the production of type 1 IFNs and many other inflammatory cytokines. Type I IFN induction is a critical component of innate immune response to several bacterial pathogens including *S. aureus* [40]. 

In a murine pulmonary *S. aureus* infection model, STING deficiency resulted in increased mortality and a higher bacterial burden in the lungs and bronchoalveolar lavage (BAL) fluid, alongside a severe destruction of lung architecture [88]. However, STING deficiency did not worsen pulmonary inflammation during the early stage of infection [88]. Another study that implicated cGAS/STING signalling in *S. aureus* infection showed that *S. aureus* infection promoted IFN-β mRNA expression and TBK1/IRF3-dependent production of IFN-β in murine macrophages [40]. Production of IFN-β has been reported to be protective against *S. aureus* infection, as IFN-β is required to turn on downstream genes important for local inflammatory responses [89]. RNA-sequencing of bone-marrow-derived macrophages (BMDMs) treated with live *S. aureus* showed that approximately 95% of the induced genes within the first four hours were mostly regulated by TLR and STING signalling pathways [39]. This study has also implicated STING, but not TLR, in the type I IFN response with live, but not killed, *S. aureus* [39]. Overall, although the studies are limited outside of mouse, cGAS/STING signalling appears to be an important player in innate immune responses to *S. aureus*.

## 3. Key Cells of the Innate Immune Response to *S. aureus* Infection

### 3.1. Keratinocytes (KCs)

KCs form the outermost barrier of the epidermis and are considered nonprofessional phagocytes, which are typically the first cells that encounter pathogens [21,90]. They recognise PAMPs via PRRs, scavenger receptors CD36, and MARCO, and subsequently signal to induce activation of transcription factors NF-κB, AP-1, and CREB, stimulating production of cytokines, chemokines, antimicrobial peptides, and inducible nitric oxide (NO) synthase [21,90,91,92]. KCs are important in immune responses to prevent *S. aureus* skin infections since they are able to phagocytose *S. aureus* and mediate caspase-1-dependent clearance through the activation of an inflammatory programmed cell death pathway, pyroptosis [93].

### 3.2. Macrophages

Macrophages are professional phagocytes and are critical for the clearance of invading pathogens from the site of infection. In macrophages, staphylococcal lipoproteins induce proinflammatory cytokines, NO, and reactive oxygen species (ROS) [94,95,96,97]. In a murine model of peritoneal infection, the particulate form of the staphylococcal cell envelope activated macrophages by inducing the production of chemotactic cytokines [98]. In *S. aureus* sepsis, increased microbial load and mortality was noticed in mice lacking macrophages [99]. Similar responses were also observed in murine airway infection models [100,101] and zebrafish [102], where loss of macrophages leads to increased *S. aureus* load and loss of normal alveolar structures. The antimicrobial peptide calprotectin present in monocytes and early macrophages successfully inhibits *S. aureus* growth in a mouse abscess model by reducing bioavailability of metal ions, starving the bacteria from essential nutrients [103,104,105].

In humans, Extracellular Vesicles (EVs) released by *S. aureus* are phagocytosed by macrophages and trigger NLRP3 inflammasome activation [106]. Another study showed that *S. aureus* infection induces a caspase-dependent pyroptotic cell death pathway in human macrophages by inhibiting mTORC1/STAT3 [107]. It is reported that upon phagocytosis by human macrophages, *S. aureus* persists in intracellular vacuoles for 3–4 days before escaping into the cytosol [108]. Treatment of human macrophages with interferon-γ at concentrations equivalent to human therapeutic doses at this stage is able to eliminate intracellular staphylococci [108].

### 3.3. Neutrophils

Neutrophils are capable of eliminating bacteria by producing nonspecific antimicrobial molecules. In a wide variety of tissues and organs, defects in neutrophil number or function result in an increased susceptibility to *S. aureus* infections [109,110,111,112]. Neutrophil priming is mediated by bacterial products including peptidoglycan and cytolytic toxins [113,114], inducing increased adhesion, phagocytosis, superoxide production, and degranulation cytotoxic granules [115]. For efficient killing and degradation, neutrophils utilise a combination of NADPH oxidase-derived reactive oxygen species (ROS), cytotoxic granule components, antimicrobial peptides, and neutrophil extracellular traps (NETs) [116,117]. In response to live *S. aureus* in humans, neutrophils can release NETs that trap the bacteria independent of ROS [118,119]. Human neutrophils are also efficient in phagocytizing biofilm-associated *S. aureus* in vitro [119,120]. However, the degree of clearance is dependent on the maturation state of the biofilm [120].

### 3.4. Dendritic Cells

DCs are the most potent antigen-presenting cell and are pivotal in the generation of adaptive immunity. Steinman and Cohn discovered the cell and applied the term “dendritic cells” based on their unique morphology [121]. Immature dendritic cells are present in all tissues including lymphoid and nonlymphoid organs [122]. Upon detecting ‘danger signals’, immature DCs are recruited to the sites of inflammation in peripheral tissues [123,124]. DCs subsequently take up antigen, migrate to lymphoid organs, and mature in the process [124]. Expression of chemokine receptors CCR6 and CCR7 also increase during maturation, which allows them to migrate to blood or lymph node to present the antigen [125,126] to prime naive T cells and initiate primary T-cell-mediated immune responses [127,128]. During maturation, DCs acquire an enhanced capacity to accumulate peptides, major histocompatibility complex (MHC) class I and II molecules, costimulatory molecules (such as CD40, CD80, and CD86), and soluble antigens (such as CD83 and DC-LAMP) [128,129].

DCs represent a heterogeneous population of cells and can be broadly divided into four categories based on surface marker expression and functional specialisation. The first two subsets, including the cross-presenting cDC1 and helper T-cell priming cDC2, comprise conventional DC (cDC) subsets [130,131] and are grouped into two categories based on their differential dependence on transcription factors for development [131]. The third DC subset includes the inflammatory monocyte-derived DCs (MoDCs), which are primarily involved in driving inflammation during infection and presenting antigen to T-cells [132]. Lastly, plasmacytoid DC (pDC) form the fourth functional group, which are involved in antiviral cytokine secretion and initiating innate immune responses [133].

While there is a body of literature focussing on the molecular mechanisms regulating innate recognition of *S. aureus* by DC, much of it is either assumed from DC receptor expression and their known ligands [134] or findings of *S. aureus* interactions with other phagocytes [119,135]. The latter tends to be obtained from either monocyte-inducible DCs or macrophage models that do not recapitulate primary DC subsets [136,137,138]. Especially lacking is literature pertaining to the role of primary cDC subsets in response to *S. aureus*, with few direct experiments investigating cDC subsets and their function either in vitro or ex vivo.

Study on BDCA1(+) myeloid DCs (mDCs) indicated that they can induce both innate and adaptive immune response against *S. aureus* [139]. In this study, BDCA1(+) mDCs phagocytosed *S. aureus* and upregulated expression of costimulatory markers MHC classes I and II, and increased production of proinflammatory cytokines and gamma interferon (IFN-γ) by CD4 and CD8 T cells [139]. Parcina and colleagues [140] examined the molecular mechanisms regulating the innate recognition of *S. aureus* by human pDC, demonstrating that activation is dependent on sensing of endosomal nucleic acids via TLR7 and TLR9 [140]. Therefore, it has been demonstrated that human pDCs can directly respond to *S. aureus* and produce tumour necrosis factor (TNF-α), IL-6, IFN-α, and upregulate CD86 expression [141]. Collectively, these observations have important implications for the immunoregulation and suppression of the host during clinical infection.

In murine models, it is clear that cDCs are important players in the recognition of *S. aureus*^143^. Indeed, cDC-deficient mice (CD11c-DTR transgenic) exhibit increased bacterial loads, accelerated mortality, and more severe pathology than controls [142]. Furthermore, *S. aureus* stimulation elicits cDC-dependent IL-12p70 [142], suggesting a possible role for cDC1 in Th1 induction. Nonetheless, these findings should be treated with caution due to the nonspecific ablation of other CD11c expressing cells, including monocytes, macrophages, NK, and B-cells in the CD11c-DTR model [143].

More recently, Richardson and colleagues [144] demonstrated a role for the *S. aureus* phenol soluble modulins in altering the Th balance in a murine infection model [144]. This study demonstrated the development of less stimulatory DCs, inducing the reduction in both Th1 and Th17 numbers, but increase in T-regs, in response to *S. aureus* lab strain USA300 expressing PSM, when compared with knockout strains [144].

In terms of DC pathogenesis, *S. aureus* is known to induce DC toxicity, with decreased numbers of murine splenic cDC1 and cDC2 during infection [144]. Recently, it has been shown that the toxicity of *S. aureus* leucocidins in monocyte-derived DCs are primarily induced through the action of pore forming leucocidin and leukocidin AB [145]. However, given the choice of monocyte-derived DCs for this research, these findings may not necessarily be reflective of primary DC subsets. Of note, *S. aureus* has also been shown to survive and replicate following internalisation by CD11c^+^ splenocytes in vitro [142], comprising a mixture of cDC, macrophages, and their precursors [131]. It is therefore clear that further work is required to understand the nuances of *S. aureus* immunopathogenesis, especially so with regard to distinct primary DC subsets.

## 4. Strategies of *S. aureus* to Evade Immune Defence Mechanisms

*S. aureus* has the capacity to survive even in the presence of an antigen-specific adaptive immune response [146]. It possesses several mechanisms to manipulate humoral or T-cell-mediated immune responses by secreting specific proteins or superantigens [146]. The remainder of this review will focus on the different immune evasion strategies used by *S. aureus* to modulate innate immune defences.

### 4.1. Inhibition of Pattern Recognition Pathways

Superantigen-like proteins (SSLs) are one of the major immune evasion molecules produced by *S. aureus* that interfere with a variety of innate immune defences [147,148]. The role of SSLs in downregulating early recognition signals through PRR and contributing to pathogenesis has been shown in a large number of studies [149,150,151,152,153]. SSL3 blocked binding of bacterial lipopeptides to the extracellular domain of TLR2 and prevented the formation of TLR2–TLR1 and TLR2–TLR6 heterodimers [153,154]. SSL3 has also been shown to block TLR2-mediated secretion of TNF-α in murine macrophages [153].

The TLRs and IL-1Rs superfamily have a conserved cytoplasmic region, known as the Toll/IL-1 receptor (TIR) domain [154,155,156]. The intracellular TIR domain can interact with a variety of TIR-containing adaptor proteins (MyD88, TIRAP, TRIF, TRAM, and TRAF6) and activate downstream signalling pathways [156,157]. The gene encoding a homologue of the human TIR domain TIR-containing protein (TirS) was identified in the mobile genetic element of ‘staphylococcal chromosomal cassettes’ (SCC) in *S. aureus* lab strain MSSA476 [156,158]. TirS in these *S. aureus* strains was localised within SCC, together with genes encoding fusidic acid resistance and genes providing resistance to methicillin. This protein exerted a specific inhibitory effect against TLR2-mediated NF-*κ*B activation, JNK phosphorylation, and production of proinflammatory cytokines [156]. TirS was reported to be present in 12% of *S. aureus* (both Methicillin-sensitive and -resistant *S. aureus*) strains, and found to downregulate the NF-*κ*B pathway through inhibition of not only TLR2, but also TLR4, TLR5, and TLR9 [159]. A comparison of pathogenicity of the MSSA476 wild-type strain vs. its isogenic *tirS* mutant in an intravenous mouse infection model revealed that presence of TirS increased the bacterial load in multiple organs [156].

### 4.2. Altering Secretion of Pro-Inflammatory Cytokines

Another staphylococcal virulence factor, adenosine synthase A (AdsA), catalyses the hydrolysis of adenosine monophosphate (AMP) to adenosine and increases the overall abundance of extracellular adenosine. Interaction of adenosine with its receptor triggers anti-inflammatory signalling cascades that cause the inhibition of platelet aggregation [160], cellular degranulation [161], IL-1α (IL-1) production, and increased production of the anti-inflammatory cytokine IL-10 [162]. Adenosine also impacts antigen presentation by decreasing the expression of MHC-II on macrophages [163] and DCs [164]. Further, adenosine reduces macrophage production of IL-12, which is a pivotal stimulus for Th1-type immune responses [165]. Thus, AdsA contributes to an overall downregulation of the innate immune response, whilst directly modulating the activation of downstream adaptive immunity.

### 4.3. Inhibition of Intracellular Degradation

*S. aureus* employs an array of mechanisms to interfere with normal host endosomal acidification, which is required for the clearance of the pathogen and the induction of an adaptive immune response. In immature bone-marrow-derived DCs, incomplete assembly of the *S. aureus* enzyme V-ATPase results in reduced acidity of host phagosomes [166]. Similarly, urease secreted by *S. aureus* increases endosomal pH by converting ammonia to urea [167]. Intracellular acidification has been shown to generate endogenous reactive oxygen species and mediate *S. aureus* cell death [168,169,170], whereas pH homeostasis mediated by urease protects cells from endogenous ROS in the presence of urea [167]. In the phagolysosomes, staphylococcal cascade proteins SodA, SodM, and KatA detoxify O_2_- into H_2_O_2_ and reduce the bacterial cell damage [171,172]. *S. aureus* also encodes accessory gene regulator (agr)-dependent factor(s) that induces autophagosome formation but prevents autophagolysosomal fusion events [173]. It has also been demonstrated that intracellular *S. aureus* within an autophagosome phosphorylates mitogen-activated protein kinase 14 (MAPK14) and autophagy-related 5 (ATG5) in murine fibroblasts, preventing autophagosome maturation and, therefore, bacterial degradation [174,175]. *S. aureus* can further escape into the cytosol from autophagosomes, where it proliferates and induces caspase-independent host cell death [173].

As elucidated for the gram-negative bacterium *Klebsiella pneumoniae* [176], *S. aureus* has also been shown to diminish macrophage efferocytosis and instead induce programmed necroptosis of neutrophils that have phagocytosed and/or been infected with *S. aureus* [177]. The secretory virulence factor alpha toxin of CA-MRSA has been implicated in impairing neutrophil efferocytosis by alveolar macrophages during lung infection [178]. This impaired clearance of neutrophils thereby delays clearance of the bacteria within the neutrophils, increases lung damage, and potentiates the capacity of antibiotic-resistant *S. aureus* to colonise the lung [178].

*S. aureus* produces a wide range of cellular and extracellular factors that helps them escape intracellular degradation and antigen presentation [179,180]. Resistance of *S. aureus* to host lysozymes is mediated by O-acetyltransferase A (OatA), which adds an acetyl group to N-acetylmuramic acid in the peptidoglycan of the cell wall [181]. The acetyl group causes steric hindrance, preventing the binding of lysozyme to peptidoglycan [182]. *S. aureus* produces PSM, which can form membrane pores in the phagosome facilitating escape from destruction and host antigen processing pathways [183,184,185]. PSMs can also activate the p38-CREB pathway in bone-marrow-derived DCs and reduces TLR2 signaling [186,187]. The *ess*-encoded virulence factor EsxA prevents host apoptosis, therefore impeding clearance, while EsxB reduces cytokine production by primary human DCs [188]. Host cell death is considered important to limit the dissemination of intracellular pathogens by revealing them to extracellular immune surveillance mechanisms [189,190]. Several studies showed that *S. aureus* can persist in human infection and serve as a reservoir for recurrent infection [191,192,193,194,195].

## 5. Development of Antibiotic Resistance and Evasion of Immune Recognition

The emergence of Penicillin-resistant *S. aureus* in the late 1940s resulted in the development of a semisynthetic penicillin (Methicillin) [196]. In the early 1960s, Methicillin-resistant *S. aureus* (MRSA) was reported and then spread worldwide. MRSA is now endemic in health care facilities in virtually all industrialised countries [197]. Community-associated MRSA (CA-MRSA) appeared in the 1990s and is currently a major problem in many countries worldwide [197,198]. Unlike healthcare-associated MRSA infections, CA-MRSA typically causes disease in healthy individuals. *S. aureus* has the capacity to acquire antibiotic resistance [199,200,201,202] by bacterial gene mutations and horizontal transfer of resistance genes from external sources [201,203,204]. Resistance to methicillin is mediated by the mecA gene carried on a distinct mobile genetic element, SCCmec [203]. mecA gene encodes a penicillin-binding protein, PBP2a, which is intrinsically resistant to inhibition by β-lactams [203,204,205]. Although SCC*mec* is central for antibiotic resistance, there is no direct evidence that SCC*mec* plays a clear role in MRSA virulence [206].

The acquisition of MRSA resistance to the last-line antibiotics vancomycin and daptomycin has been recorded over the last 15 years [207,208,209]. The mechanisms of daptomycin resistance (DAP-R) in *S. aureus* are generally thought to be associated with an overall change in the net charge of the bacterial membrane that results in the repulsion of the DAP antibiotic molecule from the cell surface [210,211]. The gene *mprF* is most consistently associated with the development of DAP-R in *S. aureus* both in vivo and in vitro [210,212,213,214]. Several studies have demonstrated that mutation in *mprF* is responsible for the cross-resistance of MRSA to DAP and Vancomycin (VCM) [215,216,217]. Other genes implicated in the development of DAP-R in *S. aureus* are those involved in phospholipid metabolism [210]. Importantly, *S. aureus* (and other Gram-positive organisms) harbor two (or more) cls genes (*cls1 and cls2*) and changes in this enzyme play a role in DAP-R by altering the pool of PG and Cardiolipin (CL) in the bacterial CM [218,219]. Another gene involved in phospholipid metabolism and DAP-R phenotype is pgsA, which encodes a CDP-diacylglycerol-glycerol-3-phosphate 3-phosphatidyltransferase [210]. Genomic analysis performed by Peleg and colleagues found that mutations in pgsA were often observed in DAP-R laboratory derivatives but not in DAP-R clinical isolates [211]. These mutations likely impair the enzymatic activity and thus decrease the overall pool of PG.

Interestingly, DAP-R clinical isolates of *S. aureus* have been shown to be significantly less virulent than susceptible parent strains following daptomycin therapy and clinical failure [220]. Cameron and colleagues [220] demonstrated that these resistant clinical isolates are not only less virulent, lending an increased rate of bacterial survival in murine challenge models, but that these strains also persist for longer in vivo than their DAP-susceptible parent strains [220].

## 6. Interplay between Antibiotic Resistance and Innate Immunity

With the mutations conferring antibiotic resistance being well-characterised, growing evidence is emerging regarding a duality in function, whereby antibiotic resistance mutations further modulate immune effector mechanisms (Figure 1).

Perhaps most well-characterised is that of the multipeptide resistance factor (MprF), which is an evolutionarily conserved protein expressed by various bacterial species including *S. aureus* [223,231,232]. The MprF protein attenuates the negative charge of anionic phospholipids in the bacterial cell membrane [233], serving to protect bacteria from the action of several host antimicrobial peptides including defensins, kinocidins, cathelicidins, and other cationic antimicrobial peptides [223,234,235]. Additionally, MprF is a flippase for Lys-PG and is required for translocation of lysinylated phospholipids to the outer leaflet of the membrane [236]. As previously discussed, gain of function mutations in MprF are associated with reduced daptomycin susceptibility. Importantly, DAP-R strains containing such point mutations have further demonstrated cross-resistance to three host cationic antimicrobial peptides [231]. While it remains unclear if these mutations are driven by host or antibiotic selective pressures (or some combination thereof), it is clear that antibiotic resistance mutations are able to simultaneously confer resistance to innate immune effectors.

Indeed, it has been demonstrated that the acquisition of specific point mutations in *cls2*, and, partly, *mprF,* correspond to a significant reduction in the secretion of proinflammatory cytokines including TNFα, interleukin-6 (IL-6), and macrophage inflammatory protein-1β (MIP-1β) as well as expression of CD80 by cDCs in response to daptomycin-resistant MRSA [225]. In vitro study of *S. aureus* also demonstrated that exposure to daptomycin alone or in combination with vancomycin or oxacillin (compared with vancomycin or oxacillin alone) led to a downregulation of TLR1, TLR2, TLR6, and TLR9 expression and dampened macrophage inflammatory response with diminished TNFα secretion and reduced synthesis of Nitric oxide (NO) [229,230]. In another in vivo study of Jiang et al., the cls2 point mutations allowed *S. aureus* to evade neutrophil chemotaxis and represented a metabolic strategy used by *S. aureus* to circumvent antibiotic and immune attack [11]. Mishra et al. found that development of DAP-R in rabbits with MRSA prosthetic joint infection is associated with resistance to host defence cationic peptides (HDP-R) [232].

Aside from directly interfering with innate effectors, *S. aureus* has the remarkable capacity to also accumulate antibiotic resistance mutations that simultaneously mask ligands of critical innate sensors. Increased biofilm formation capacity in hospitalised patients by MRSA compared with Methicillin Sensitive *S. aureus* (MSSA) strains is considered an important virulence factor influencing its persistence in both the environment and the host [234]. In addition, biofilms of MRSA attenuate the expression of IL-1β, TNF-α, CXCL2, and CCL2, circumventing TLR2 and TLR9 recognition pathways and macrophage invasion into the biofilm [235].

Indeed, it has been shown that specific mutations in the 23S rRNA of *S. aureus* confer resistance to macrolide, lincosamide, and streptogramin group antibiotics, whilst simultaneously preventing detection through TLR13 [224]. TLR13 is not present in humans, and whether these mutations are also able to evade human recognition of *S. aureus* 23S rRNA remain to be confirmed. The notion that bacteria can change throughout the course of infection to evade innate immune detection is indeed novel, and these findings set an important precedent for further investigation.

## 7. Concluding Remarks

The innate immune response is the key to the prevention of infection and generation of adaptive immunity. Initial sensing of bacteria through PRR initiates the inflammatory cascade to release cytokines, antimicrobial peptides, and recruit antigen presenting DCs to further activate adaptive immune responses. However, *S. aureus* possesses a large repertoire of virulence factors including antibiotic resistance genes that can lead to suboptimal innate immune responses, opening a window to establish and sustain infection. The precise mechanisms defining how antibiotic resistance mutations affect the bacterial isolate’s ability to evade innate immune activation is of clear importance. Understanding how resistant *S. aureus* modulates innate cells, including DCs, to escape recognition is critical to restore immunogenic potential and to allow recognition and clearance.

## Figures and Tables

**Figure 1 antibiotics-11-00542-f001:**
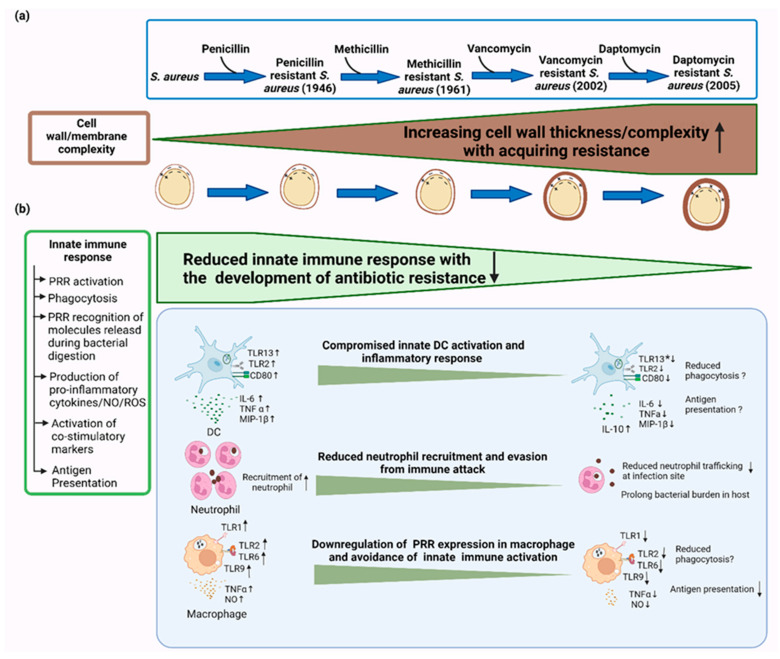
Schematic presentation of the changes in the innate immune response coincident with the development of antibiotic resistance in *S. aureus*. (**a**) Changes in membrane phospholipid composition with the evolution of antibiotic resistance in *S.*
*aureus.* Emergence of MRSA with reduced susceptibility to VCM and DAP has been associated with gain-of-function, in which lysinylation of PG is increased [217,221]. This positively charged L-PG increases net positive charge on the CM and reduced affinity for VCM and DAP [221]. Increased cell wall thickening is also suggested to cause ineffective binding of DAP/VCM to the CM [218,219]. (**b**) In the innate immune response, *S.*
*aureus* is recognised by PRR of immune cells and phagocytosed, leading to the activation of intracellular PRRs and upregulation of costimulatory markers [222]. Cytokines produced by innate immune cells further mobilise recruitment of immune cells for pathogen clearance [222]. Among the innate immune cells, Antigen Presenting cells (APCs) induce the activation of adaptive immunity through antigen presentation [223]. However, antibiotic resistant *S. aureus* can evade innate immune detection, thus preventing activation. Notably, antibiotic-resistant *S. aureus* can modulate DC activation by avoiding recognition via TLR2 and 13 [159,224]. In response to DAP-R MRSA, DCs produced reduced proinflammatory cytokines including TNFα, IL-6, and MIP-1β as well as decreased CD80 expression compared with DAP-sensitive MRSA [225]. MRSA can also suppress DC activation by inducing the production of immunosuppressive cytokine IL-10 [226]. Neutrophils are one of the most fundamental host innate immune effectors against *S. aureus* [227]. During infection, neutrophils are initially recruited to the site of infection, a process termed chemotaxis, followed by adhesion and phagocytosis of *S. aureus* [228]. On the other hand, mutation in membrane phospholipid biosynthesis gene (cls_2_) conferring DAP resistance allowed *S. aureus* to evade neutrophil chemotaxis and overcome the immune attack [11]. Macrophages, which are also professional phagocytes, can also successfully control and degrade *S. aureus* using a range of mechanisms, including TLR activation, upregulation of proinflammatory cytokines, and generation of reactive ROS and NO [135]. However, the development of antibiotic resistance in *S. aureus* led to a downregulation of TLR1, TLR2, TLR6, and TLR9 expression in macrophages and dampened TNFα secretion and NO synthesis, allowing them to survive and escape macrophages [229,230].

**Table 1 antibiotics-11-00542-t001:** PRRs involved in *S. aureus* recognition.

PRR Family	Receptor	Species		Cellular Location	PAMP	Signal Adapter	References
		Human	Mouse				
TLR	TLR1/TLR2 heterodimer	+	+	Plasma Membrane	Triacyl lipoproteins	MyD88/TIRAP	[18,19,20,21,22]
	TLR2/TLR6 heterodimer	+	+	Plasma Membrane	Diacyl lipoprotein, Lipopeptide, Peptidoglycan, Phenol soluble modulin (PSM)	MyD88/TIRAP	[18,19,20]
	TLR7	+	+	Endosome	tRNA	MyD88	[23,24]
	TLR8	+	−	Endosome	ssRNA	MyD88	[25,26]
	TLR9	+	+	Endosome	Unmethylated CpG DNA	MyD88	[23,27]
	TLR13	−	+	Endosome	23s RNA	MyD88	[23]
CLR	MBL	+	+	Plasma Membrane	Teichoic acid		[28,29,30]
NLR	NOD2	+	+	Cytoplasm	Peptidoglycan, α, β-, and γ- hemolysins; Panton–Valentine leukocidin (PVL); Leukocidin A/B; Acylated lipopeptides	RIP2, MAVS	[31,32,33]
RLRs	RIG-I	+	+	Cytoplasm	Cytoplasmic RNA	MAVS	[34,35]
CDS	cGAS STING	+	+	Cytoplasm	Double stranded DNA, Cyclic dinucleotide	STING	[36,37,38,39,40]

CDS, Cytoplasmic DNA sensors; MyD88, Myeloid differentiation factor 88; TIRAP, Toll–interleukin 1 receptor (TIR) domain-containing adapter protein; CpG, cytosine-phosphate-guanosine; RIP2, Receptor-interacting-serine/threonine-protein kinase 2; MAVS, Mitochondrial antiviral-signalling protein.

## Data Availability

Not applicable.

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
