# Peer review of "The Influence of Antibiotic Resistance on Innate Immune Responses to Staphylococcus aureus Infection"

_antibiotics, 2022, doi:10.3390/antibiotics11050542_

Round 1
Reviewer 1 Report
This is an interesting review the strategies used by Staphylococcus aureus for circumvent antibiotic and innate immune responses, especially the interaction between S. aureus and dendritic cells, focusing on how this relationship is disrupted as antibiotic resistance develops. The manuscript is well written and well summarizes the individual elements of the widely described topic.
Maybe the section devoted to the main topic of manuscript should be revised by introducing more references related to the main topic. Are there data on the influence of S. aureus strains (other than MRSA, VRSA or DAP-R) resistant to antibiotics on the mechanisms of innate immunity to infection?
Moreover, I propose to pay more attention to the references: items 147 and 148 contain the same publication, as do items 153 and 158, as well as 226 and 237. In reference 169 remove capital letters. I recommend checking all the references carefully.
Author Response
1.Are there data on the influence of S. aureus strains (other than MRSA, VRSA or DAP-R) resistant to antibiotics on the mechanisms of innate immunity to infection? No, these strains are the major antibiotic-resistant strains that have developed in response to antibiotic use- see for eg: https://doi.org/10.1093/femsre/fux007
2. Thanks for pointing out duplicate references. References have been checked, duplicates removed and unusual symbols that had been incorporated in the endnote conversion have been amended.
Reviewer 2 Report
Submitted manuscript Antibiotics-1673297 “The influence of antibiotic resistance on innate immune responses to Staphylococcus aureus infection” authored by Nazneen Jahan, Timothy Patton, and Meredith O’Keeffe is very well composed manuscript.
I appreciate that author has done the intensive literature search and touch most of the components with relevant citations, Table 1 is literally summarizing the major portion of paper. Although, some of the place’s use of literature ought to do cautiously; like line number 130 use of word “primarily” would be much appropriate but the cited paper reference no. 50 is not relevant.
I found that most of the aspects of the S. aureus evasion mechanism of host immune cells and intracellular machinery has been covered here. However, in section 3.2 the efferocytic clearance of apoptotic cells and section 4.3 inhibition of intracellular degradation by hijacking the apoptotic pathway by bacteria are missing. I will suggest to author/s that include the discussion of these aspect as well. You can find it these aspects in Jondle et al 2018 and you should cite as well. https://doi.org/10.1371/journal.ppat.1007338
Likewise, Line number 458 “a) Changes in membrane phospholipid composition” here the role of membrane protein flippases and scramblases need to mention somewhere as described in Jondle et al 2018.
Figure 1 legend should be properly formatted.
Line number 487 “* Not expressed on human cells.”?
Similarly, some places one line has one information but supported by 3-4 citations. I encourage that use only most relevant and recent review as accurate as possible. Consecutively, review will look like more technically efficient.
Please take care of above mentioned typographical and formatting mistakes. Overall, the submitted manuscript also warrants some minor language corrections and proofreading as per the suggestions.
All the best.
Author Response
1. .Re referencing: ....line number 130 use of word “primarily” would be much appropriate but the cited paper reference no. 50 is not relevant.
We take your point with this citation that had looked only at myeloid CLRs. We have included a more general Reference for CLRs (new Ref 50) which outlines their different cellular expression and the fact that many are expressed by myeloid cells.
New text is inserted at lines 130-131:
C-type lectin receptors (CLRs) are a heterogenous group of transmembrane proteins, many of which are expressed on myeloid cells [50].
2. Discuss efferocytic clearance
We have included new text lines 401-08
As elucidated for the gram negative bacterium Klebsiella pneumoniae [179], S. aureus has also been shown to diminish macrophage efferocytosis and instead induce programmed necroptosis of neutrophils that have phagocytosed and/or been infected with S. aureus [180]. The secretory virulence factor alpha toxin of CA-MRSA has been implicated in impairing neutrophil efferocytosis by alveolar macrophages during lung infection [181]. This impaired clearance of neutrophils thereby delays clearance of the bacteria within the neutrophils, increases lung damage, and potentiates the capacity of antibiotic resistant S. aureus to colonize the lung [181].
3. Line number 458 “a) Changes in membrane phospholipid composition” here the role of membrane protein flippases and scramblases need to mention somewhere as described in Jondle et al 201
Discussed at line 503-05:
Additionally, MprF is a flippase for Lys-PG and is required for translocation of lysinylated phospholipids to the outer leaflet of the membrane [237].
4. Figure 1 legend is now formatted and the 'orphan' line below it now corrcted- thank-you. (line 482)
5. Referencing- cite reviews rather than individual manuscripts: We think it is best practice to provide the reader with the original data from different labs where available.
Have gone through for additional proofreading and updated (highlighted in yellow).
Thank-you